# Galectins as Checkpoints of the Immune System in Cancers, Their Clinical Relevance, and Implication in Clinical Trials

**DOI:** 10.3390/biom10050750

**Published:** 2020-05-12

**Authors:** Daniel Compagno, Carolina Tiraboschi, José Daniel Garcia, Yorfer Rondón, Enrique Corapi, Carla Velazquez, Diego José Laderach

**Affiliations:** 1Molecular and Functional Glyco-Oncology Laboratory, IQUIBICEN-CONICET-UBA, Ciudad Autónoma de Buenos Aires C1428EGA, Argentina; carotiraboschi@gmail.com (C.T.); Jdanielgarcia07@gmail.com (J.D.G.); Ferrg09@gmail.com (Y.R.); ecorapi@gmail.com (E.C.); carvlzqz@gmail.com (C.V.); 2Departamento de Química Biológica, Facultad de Ciencias Exactas y Naturales, Universidad de Buenos Aires, Ciudad Autónoma de Buenos Aires C1428EGA, Argentina; 3Facultad de Biotecnología y Biología Molecular, Facultad de Farmacia, Universidad Nacional de la Plata, La Plata 1900, Provincia de Buenos Aires, Argentina; 4Departamento de Ciencias Básicas, Universidad Nacional de Lujan, Lujan 6700, Provincia de Buenos Aires, Argentina

**Keywords:** galectins, carbohydrates, galectin

## Abstract

Galectins are small proteins with pleiotropic functions, which depend on both their lectin (glycan recognition) and non-lectin (recognition of other biomolecules besides glycans) interactions. Currently, 15 members of this family have been described in mammals, each with its structural and ligand recognition particularities. The galectin/ligand interaction translates into a plethora of biological functions that are particular for each cell/tissue type. In this sense, the cells of the immune system are highly sensitive to the action of these small and essential proteins. While galectins play central roles in tumor progression, they are also excellent negative regulators (checkpoints) of the immune cell functions, participating in the creation of a microenvironment that promotes tumor escape. This review aims to give an updated view on how galectins control the tumor’s immune attack depending on the tumor microenvironment, because determining which galectins are essential and the role they play will help to develop future clinical trials and benefit patients with incurable cancer.

## 1. Introduction

Worldwide, cancer is a major public health problem. It has become the leading cause of death in developed countries, while it shows a continuous progression in developing countries [1]. It is, therefore, essential to understand the molecular processes involved in the emergence, amplification, and potential therapeutic regression of transformed cells.

The immune system participates in each of the three aforementioned stages that determine the final result of the disease. On the basis of epidemiological studies in immunodeficient patients, plentiful evidence supports the effectiveness of the immune system to eliminate transformed cells, even if they belong to the body itself. The most studied case is that of immunosuppressive treatments after transplants, where the development of cancer is much more frequent than in healthy subjects [2,3]. Moreover, the active role of the immune system controlling tumors is also supported by clinical observations that correlate spontaneous tumor regression with autoimmunity (i.e., melanoma regression in the context of vitiligo; one of its initial descriptions by [4]). Moreover, various experimental models support the concept of cancer immunoediting, in which the immune pressure shapes tumor immunogenicity [5]. Altogether, evidence indicates that the immune system does not ignore transformed cells and participates in the elimination of tumors. Tumor immune attack occurs very efficiently at the early stages of cell transformation, in which the tumor burden is minimal and probably with a high frequency during the life of an individual.

However, in some cases, transformed cells bypass immune recognition and prevent rejection. If this occurs, tumors enter the immune escape phase, being much less immunogenic [6]. Various mechanisms are involved in this escape phase. First, intrinsic tumor changes are required, such as a reduced expression of biomolecules involved in immune recognition (major histocompatibility complex, co-stimulation, and so on). The accumulation of metabolic and immunosuppressive factors in the tumor microenvironment is also observed, including indoleamine-2,3-dioxygenase (IDO), arginase, or TGFβ1, among other soluble factors. Second, tumors activate immune checkpoints (ICPs) as an effective means to deactivate immune rejection (Figure 1). In fact, under normal physiological conditions, ICPs are crucial for the maintenance of self-tolerance (the prevention of autoimmunity), as well as to protect tissues from damage when the immune system is responding to pathogenic infection (Figure 1). Tumors deregulate these same molecular pathways as a critical immune escape mechanism.

Targeting ICP by inhibitory immune checkpoint molecules has dramatically changed treatment paradigms in medical oncology. In clinics, ICP blocking releases the potential of the antitumor immune response, thereby transforming the therapy of human cancer. Among them, PD-1 (also known as CD279) interacting with PD-L1 (B7-H1 or CD274) or PD-L2 (B7-DC, CD273), and CTLA-4 (also known as CD152) interacting with CD80 (B7.1) or CD86 (B7.2), are two molecular axes extensively tested in clinical trials. Their extensive evaluation is supported by our current knowledge on the implicated signaling cascades, their expression kinetics, their favorite cell targets, and the anatomical location for their optimal function [7]. However, clinical results on the intervention in these molecular axes (antibody-mediated blockade) have shown partial results [8,9], and some cancers like prostate cancer (PCa) are refractory diseases. Altogether, it demonstrates the need to explore other negative checkpoints. Among them, B7-H3 (CD276); lymphocyte activation gene 3 (LAG-3, CD223); T cell immunoglobulin and mucin-3 (TIM-3); KIR molecules in natural killer (NK) cells; V-domain Ig-containing suppressor of T cell activation (VISTA); the T cell ITIM domain (TIGIT); CD112R; CD39; CD73; and adenosine A2A receptor are alternative targets to be taken into account (reviewed in depth elsewhere [10,11,12]) (Figure 1).

Currently, each of these types of interactions is being investigated. It is interesting to highlight that, along with the axis PD-1/PD-L1 y CTLA-4/B7, these negative ICPs act on various stages of the anti-tumor response, including activation and proliferation in secondary lymphoid tissues, lymphocyte trafficking to sites of antigen expression, the execution of direct effector functions, the provision of help for immune cells (through cytokines and membrane ligands), the metabolic control at the clonal expansion and effector phases, the induction of a deactivated/exhausted state in lymphocytes, the functional promotion of regulatory T cells, the activation of innate cells that has an indirect effect on adaptive immune responses, and finally the regulation of antibody production by B lymphocytes. Most of these actions are performed through receptor-mediated intracellular signaling in lymphocytes. However, there are some effects described as occurring through “signaling independent ways”, as in the case of CTLA-4, which depletes CD80 and CD86 from the antigen-presenting cells and therefore has an impact on antigen presentation [13]. The use of blocking antibodies may prevent both types of immune checkpoint actions.

In this review, we propose that galectins (Gals) are additional negative checkpoints of the immune response in the context of a tumor microenvironment for various reasons. First, their production is differential between physiological tissues and tumors [14,15]. Second, Gals can modulate anti-tumor immune responses by inducing T cell death [16], the functional deactivation of lymphocytes [17,18,19], the activation of suppressor mechanisms [20,21,22], and/or the induction of regulatory dendritic cells [23] that in situ sense the tumor and transport such information toward the draining lymph nodes to prime the anti-tumor immune response. However, not only adaptive immunity is regulated by Gals. It has been reported that polymorphonuclear neutrophils [24], macrophages and myeloid cells [25,26,27], NK cells [28,29,30], B cells [31,32], endothelial cells [33], and platelets [34], among other cells, may be the target of functional modulation by Gals. Regulation on innate cells contributes in situ to create a microenvironment permissible for tumor growth. Most of these biological effects are mediated by the interaction of Gals with their ligands fully exposed on the surface of the tumor cells (glycans in the form of biological glycoconjugates). This interaction also induces intracellular signals in immune/hematological cells.

Additionally, we will discuss other non-lectin mediated (glycan-independent) functions that have also been ascribed to Gals in their regulation of anti-tumor immune responses. Last, but not least, as the initial immune activation and expansion events occur in the draining lymph nodes, the tumor secretes and dispatches Gals to these distant anatomical locations inside microvesicles [35,36,37]. Gals can also act in an indirect manner, preventing in situ functions of dendritic cells that sense the tumor and should carry activating anti-tumor signals to the lymph nodes. In this way, Gals produced by tumors have a direct and negative effect on lymphocyte primo-activation as well as on the effector phase of cytotoxic T lymphocytes. While the expression of galectins in a tumor environment is well documented, this review will present the most recent information about the effects of main Gals as regulators of the immune response and their prognosis potential for successful immunotherapy (focusing on reports from the last five years). Finally, we will analyze the main clinical trial results highlighting the idea that co-targeting galectins in combination with other immunotherapy (e.g., ICP targeting and/or therapeutic vaccination) may improve and solve several cases and benefit cancer patients.

## 2. Galectins as Regulators of Immune Responses

Galectins (S-type lectins) are a family of evolutionarily conserved glycan-binding proteins characterized by their affinity for N-acetyllactosamine sequences. The latter residues can be displayed on cell surface glycoconjugates or expressed intracellularly as well as in the extracellular matrix [14,38]. Gals are involved in the regulation of different cellular processes, among which are cell adhesion and migration; cell differentiation; gene transcription and RNA splicing; cell cycle; and apoptosis [39]. Their expression is altered during several pathological conditions, including cancer and autoimmunity [14,15]. Despite their multiple functions in cancer cell behavior (e.g., tumorigenesis, metastasis, and angiogenesis) [33,40,41,42,43], Gals also have essential immunomodulatory functions that regulate both innate and adaptive anti-tumor immunity [44]. These functions are strongly dependent on the cell context, and the concomitant expression of specific Gals and related carbohydrate ligands [39]. Indeed, Gals and glyco-ligands both could serve as useful diagnostic biomarkers in cancer. In most cases, the expression of galectins in the tumor microenvironment predicts a poor clinical outcome [14,45]. Among the 11 galectins identified in humans, Gals-1, -3, and -9 have been extensively studied in different fields including cell biology and immunology. Moreover their functions were closely linked to cancer biology such as enhancing tumorigenicity, regulating tumor cell growth or apoptosis, modulating cell migration, and promoting tumor immune escape.

### 2.1. Galectin-1

Galectin-1 is one of the most representative members of the prototype group. This galectin is only active with a dimeric structure. Interestingly, tumors have an ideal oxidative microenvironment indispensable for Gal-1 dimerization. The first work showing the link between Gal-1 and the immune system was published by Baum L. and coll. in 1995 [16] and reviewed earlier in [46]. Another pioneering study demonstrated that inhibiting Gal-1 gene expression in tumor cells in vivo promotes tumor rejection and stimulates the generation of a tumor-specific T cell-mediated response in syngeneic mice [47]. More recently, it demonstrated the link between Gal-1 and tumor neoangiogenesis [48,49]. Moreover, the targeting of Gal-1 inhibited pancreatic ductal adenocarcinoma (PDA) progression by modulating tumor–stroma crosstalk [50]. Likewise, another group showed that the level of tumor Gal-1 was inversely correlated with treatment response with anti-ICP and survival in patients with head and neck cancer (HNC) [51]. We can imagine a scenario in which the angiogenesis normalization induced by Gal-1 blockade enhances drug access to tumors, resulting in important clinical benefits for patients. In addition, targeting Gal-1 in patients also favors T cell infiltration, leading to reduced metastasis rates as a secondary consequence. Indeed, Gal-1 from tumor-derived exosomes appears to be an essential protein that induces CD8+ T cell suppressors by loss of CD27/CD28 co-stimulation and attenuation of IFN-γ production. These actions of Gal-1 thus prevent T cell anti-tumor functions in HNC models [37]. All of these arguments support Gal-1 as a negative prognosis factor in different types of cancers and reveal the importance of evaluating conditioning strategies as an exciting opportunity to shape tumor fate by targeting this lectin in the tumor microenvironment. In a recent study, we assessed how hemin, a pharmacologic inducer of heme oxygenase-1 (HO-1), has an impact on prostate cancer development in vivo [52]. We demonstrated that tumors from hemin-conditioned mice showed reduced expression of Gal-1, resulting in a persistent remodeling of the microenvironment (immune and vascular systems). We also described a subset of prostate cancer patient-derived xenografts and prostate cancer patient samples with mild HO-1 and low Gal-1 expression levels. These results are an example of strategies by which Gal-1 can be modulated in the microenvironment, and thus impact tumor progression.

Analysis of the current bibliography reveals a main interest in the immune-regulatory effects of Gal-1 produced by tumor cells. However, additional sources of this galectin may also be evaluated as a modulator of the anti-tumor immune responses. Using a prostate cancer preclinical model, we demonstrated that Gal-1 is expressed by T lymphocytes at a much lower level than prostate tumor cells, but has an essential role in immune tolerance. Specific Gal-1 inhibition only in T cells is enough to potentiate anti-tumor immune responses [53]. We also demonstrated that endogenous Gal-1 in CD4+ (but mainly in CD8+ T cells) acts as a negative regulator of anti-tumor immunity, and concluded that prostate tumors require Gal-1 in lymphocytes to evade immune responses. This report lays the foundation for an original immunotherapy strategy for prostate cancer targeting Gal-1 not in the tumor cell, but in tumor-associated T lymphocytes, highlighting the necessity to identify which cells within the tumor environment express functionally active galectins.

### 2.2. Galectin-3

Galectin-3 is the unique chimeric member of the galectin family and can form a pentameric structure. Since the first original work in 1989 showing that Mac-2 antigen, actually known as Gal-3, binds to IgE from basophilic leukemia cells, a large number of subsequent studies report that Gal-3 functions in other types of tumors (recently reviewed in [54,55]). 

From more recently published data, Gal-3 has been defined as a guardian of the tumor microenvironment [56]. In the last five years, reports showed Gal-3 to exert a broad spectrum of functions, which differ according to its intra- or extracellular localization. Galectin-3 has unique structural properties among all galectins, allowing it to be the only chimeric member of the family. Indeed, it has a C-terminal composed by the carbohydrate recognition domain (CRD) and a nuclear transporter-binding domain in its distal portion with nuclear export property [57,58]. At the other end, the long N-terminal tail (120 residues) contains a stretch with two potential phosphorylation sites (Ser6 and Ser12) followed by a region containing several tandem repeats of (PGAYPG) segments. This glycine and proline-rich domain is involved in the ability of Gal-3 to oligomerize with other Gal-3 molecules or to establish protein–protein interactions with distinct proteins. Moreover, residues in the first portion of the N-terminal tail can interact with the CRD by transient contacts and control the subcellular localization of Gal-3, promoting the switch from nuclear to cytoplasmic form to allow anti- or pro-apoptotic functions of this galectin, respectively [59]. Aside from the polysaccharide binding site on the CRD of Gal-1 and -3, Miller et al. also found that rhamnogalacturonan polysaccharide (RG-I-4) could bind relatively strongly to the N-terminal end of Gal-3; this strong binding occurring kinetically slowly is most likely associated with proline cis trans isomerization [60]. It is so far evident that interfering with the unique structure of Gal-3, which is dependent on this long N-terminal domain, could control not only the lectin properties of the CRD, but also the capacity of its N terminal tail to bind polysaccharides. In addition, hindering the flexibility of the N-terminal tail of Gal-3, which promotes the interaction with the CRD, could also control other functions such as anti- or pro-apoptotic properties interfering with the nuclear/cytoplasm switch of this galectin.

Notably, Gal-3 emerged as a multifunctional protein by inducing monocyte-macrophage differentiation. Moreover, Gal-3 could interfere with dendritic cell fate decisions, regulating apoptosis on T lymphocytes and inhibiting B-lymphocyte differentiation into immunoglobulin secreting plasma cells [61]. All of these immunomodulatory effects are critical in immune-inflammatory responses. The capability of Gal-3 to bind activated antigen-committed CD8+ T cells is only possible in the tumor microenvironment through Gal-3 binding to LAG-3, and thus LAG-3 expression is necessary for galectin-3-mediated suppression of CD8+ T cells in vitro. Moreover, Gal-3-deficient mice exhibit improved CD8+ T-cell effector function and increased the expression of several inflammatory genes. Lastly, Gal-3-deficient mice also have elevated levels of circulating plasmacytoid dendritic cells, which could be superior to conventional dendritic cells in activating CD8+ T cells. These results suggest that inhibiting Gal-3, in conjunction with CD8+ T-cell-directed immunotherapies, should enhance the tumor-specific immune response.

Mordoh‘s group was the first to demonstrate that Gal-3, not Gal-1, correlates with apoptosis of tumor-associated lymphocytes in human melanoma biopsies [62], suggesting that a particular galectin could have essential roles in some cancers, but not in all of them. Several years later, another investigation group confirmed these data and revealed two galectin members (e.g., Gal-3 and -9) as being between the four most relevant parameters associated with more prolonged survival in metastatic melanoma patients [63]. Indeed, a comprehensive study of metastases of 73 stage IV melanoma patients highlighted that four parameters are associated with better response to adoptive cell transfer (ACT) as immunotherapy, including a higher number of CD8+ T cells, a high M1/M2 macrophage ratio, the presence of Gal-9 expressing dendritic cells (DCs), and the increase of the Gal-3 expression by tumor cells. In conclusion, while the presence of at least three of these parameters constitutes an independent favorable prognostic factor for long-term survival, patients that displayed this four-parameter signature were found exclusively among those who responded to ACT with sustained clinical benefit. Interestingly, the expression of Gal-1 and -3 was only different in the long-term survival group. However, differences in Gal-3 expression between patients with medium or long-term survival were not statistically significant, suggesting that higher expression of Gal-3 is not enough to improve overall survival (OS) in ACT patients, but should increase the OS in combination with or as a consequence of other parameters.

The precise role of Gal-3 in driving the inflammatory process in autoimmune or immune-mediated disorders revealed a differential role of endogenous and exogenous Gal-3 [61], similar to Gal-1 in prostate cancer [53]. More than a decade ago, Califice and coll. were pioneers in demonstrating the dual activities of Gal-3 depending on its subcellular localization [64]. For this reason, the mode of action by which Gal-3 acts on monocyte–macrophage differentiation, dendritic cell fate decision, regulation of apoptosis on T lymphocytes, and inhibition of B-lymphocyte differentiation into immunoglobulin secreting plasma cells was not reproduced by the use of recombinant and exogenous Gal-3. Indeed, the use of recombinant Gal-3 trying to replicate its endogenous actions did not induce any similar effect [61]. These results highlighted the complexity in understanding galectins’ functions in cancer development, and the need to identify not only in which cells each galectin is expressed, but also in which subcellular compartment their expression occurs.

### 2.3. Galectin-9/TIM-3

Gal-9 is one of the representatives of the tandem-repeat isoforms with two carbohydrate recognition domains. Gal-9 binds to the T cell immunoglobulin and mucin domain-containing protein 3 (TIM-3), which is expressed at the surface of terminally differentiated T cells. As shown in viral infections, Gal-9 induces either apoptosis or suppression of T cell effector functions via engagement with its receptor TIM-3 [65]. In agreement, Gal-9 knockout mice mount a more robust and vigorous virus-specific immune response, resulting in rapid viral clearance. With more than 150 publications per year in the last three years, the Gal-9/TIM-3 interaction is one of the most studied to be responsible for the immune tolerance in cancer. The Gal-9/TIM-3 pathway is functional in a wide range of human cancer cells [66]. Thus, it is important to ascertain whether Gal-9 has therapeutic potential in human diseases, including cancer, by influencing T cell immunity and inducing apoptosis of specific T-cell subpopulations like other galectins do. This concept is well documented and will be detailed in the following sections.

### 2.4. Other Galectins

Besides Gals-1, -3, and -9 (the latter mainly through its interaction with TIM-3), other galectins may have a role in tumor development. A transcriptome meta-analysis of cervical cancer cells after ectopic Gal-7 expression demonstrated the regulation of molecular networks that are involved in several cancer hallmarks, such as metabolism, growth control, invasion, apoptosis, and control of the immune response [67]. This result suggests that Gal-7 plays a role in the tumor microenvironment and immune surveillance, at least in cervical cancer. In the near future, we will probably have a more complete knowledge about the participation of other members of the galectin family in the process of tumor immune escape. 

In conclusion, galectins are factors implicated in the control of anti-tumor immune responses. However, the effort should now be directed toward a clear identification of whether tumor cells or other cells within the tumor microenvironment represent the leading source for each galectin, and the final goal is to develop specific tools to interfere with their action specifically in these cells.

## 3. Role of Galectins on Immune Cells in Cancers

Previous arguments indicate that galectins are critical molecular players in the complex process of cancer immunoediting, and particularly as tumor immune escape mechanisms. A complexity that persists today refers to the identification of the primary cellular sources for each galectin inside the tumor microenvironment (Table 1, Table 2, Table 3 and Table 4). This information is relevant, given that it defines the strategies to regulate their expression. In this sense, the three principal members (Gals-1, -3, and -9) are expressed by CD163+ macrophages. Gal-1 and -3 are poorly expressed by T cells, and the former is found in fibroblasts in head and neck squamous cervical cancer (HNSCC) [68] and prostate cancer microenvironments [53]. However, low levels of expression of a galectin in a cell type do not indicate that this member does not regulate essential functions in these cells, as we have recently reported in the case of Gal-1 in lymphocytes [53]. This concept makes the task more complicated and demonstrates that functional studies are required to have a complete picture of the role of Gals in the tumor microenvironment. At present, this information is limited. It is thus essential to study each cancer microenvironment to identify where and which immune partner expresses each of the galectins (e.g., Gal-1, -3, and -9). It is also essential to define their potential ligands and functional implications, and then develop strategies to interfere efficiently with their tumor-escape functions (Table 1, Table 2, Table 3, Table 4 and Table 5). Finally, all this knowledge will facilitate the evaluation of their clinical significance.

### 3.1. Macrophages and Myeloid Cells

Macrophages play an essential role in the immune response and the maintenance of tissue homeostasis. It is well known that many tumors recruit monocytes from the circulation and influence their differentiation, mainly into suppressive M2-like subsets. While M1-type macrophages are associated with anti-bacterial responses, polarization to M2-type differentiation intimates a variety of biological responses, including tissue repair, immunosuppression, promotion of tumor angiogenesis, and metastasis. With this concept in mind, it is interesting to note that colon cancer-derived conditioned medium induces Gal-3 expression by tumor cells, and can actively influence the phenotype of monocytes and switch their differentiation into a mixed population of M1 and M2 cells [88]. Gal-3 is also highly expressed within the tumor microenvironment of aggressive cancers, and whose expression correlates with poor survival, particularly in patients with NSCLC [89]. A preclinical study in two NSCLC mouse models showed that the combination of an orally active Gal-3 antagonist (GB1107) with blockade of PD-L1 boosts tumor immune infiltration, reducing lung adenocarcinoma growth and blocking metastasis [90]. Therefore, GB1107 augments the response to immune checkpoint inhibitors. Elegant experiments indicate that macrophages were a significant driver of this anti-tumor response. Indeed, oral administration of GB1107 increased tumor M1 macrophage polarization and CD8+ T-cell infiltration, two essential parameters implicated in the anti-tumor effect.

Gal-9 is another galectin that enhances the programming of tolerogenic macrophages. Indeed, Gal-9 modulates immunity by promoting M2 macrophage differentiation, impacting the survival of patients with metastatic melanoma [82]. The comparison of Gal-9/TIM-3 pathways in healthy and malignant myeloid cells reveals the relevance of this pathway for this type of cell. Indeed, TIM-3 expression was significantly higher in primary human acute myeloid leukemia (AML) blasts, and may thus serve as a possible target for AML therapy. The subcellular localization of TIM-3 was also different; while TIM-3 was distributed mainly on the surface of primary AML cells, healthy leukocytes showed an intracellular expression. In primary human AML blasts, both TIM-3-agonistic antibody and Gal-9 (as the natural TIM-3 ligand) significantly upregulated the mTOR pathway, enhancing the accumulation of pro-tumoral factors including hypoxia-inducible factor 1 alpha (HIF-1α) and the secretion of VEGF and TNF-α [97]. Moreover, the TIM-3/Gal-9 pathway promotes leukemia stem cells’ survival, inducing the expansion of myeloid-derived suppressor cells (MDSCs) and differentiation into tumor-associated macrophages [83].

Gal-9 also binds to Dectin-1, a natural killer (NK)-cell-receptor-like C-type lectin involved in innate immune responses to fungal pathogens. In pancreatic adenocarcinoma, Gal-9 binds to Dectin-1 on macrophages, resulting in its tolerogenic programming. These polarizations of macrophages toward a pro-tumoral M2 phenotype have a profound impact on the T-cell cytokine secretion profile, promoting the tumor immune escape [81]. Finally, glioma-derived Gal-1 regulates innate and adaptative anti-tumor immunity [28]. Using an orthotopic GL261 mouse high-grade glioma model, the authors nicely demonstrated that tumor-derived, but not host-derived, Gal-1 inhibition significantly prolonged the survival of glioma-bearing mice after a dendritic cell (DC)-based immunotherapy. A significant decrease in the number of brain-infiltrating macrophages and myeloid-derived suppressor cells (MDSC) is associated with this anti-tumor effect, demonstrating a pivotal role of Gal-1-expressing tumor cells in cancer immune evasion.

### 3.2. Dendritic Cells

During the two last decades, the role of galectins on DC differentiation or function was richly documented in infectious models, analyzing the function of endogenous and exogenous (including DC-derived exosomas) galectins. Gal-1 seems to have opposite effects on DC behavior, depending on the experimental conditions. On the one hand, Gal-1 stimulates the secretion of proinflammatory cytokines [72] or the maturation of DC [71]. Moreover, this particular galectin endows DC with tolerogenic potential mediated through an immunoregulatory circuit involving interleukins 27 and 10 [23]. Furthermore, several examples show that other galectins regulate DC behavior, including Gal-9 for DC maturation [100], and Gal-3 controlling adhesion and migration properties of these cells [79]. Remarkably, most of these biological effects depend on the glycophenotype at the surface of DC, demonstrating the requirement of outside-in signaling [101]. Mass spectrometry of human DC revealed a specific interaction between CD69 and Gal-1, but not with Gal-3 or Gal-7 [69]. In addition to CD69, Gal-1 binds to surface CD45 and CD43 in DC, interactions that also impact their activation and migration properties [70].

Antigen-presenting cell (APC) subsets, including Kupffer cells (KCs), myeloid dendritic cells (DCs), and plasmacytoid DCs in HCC, express different levels of Gal-9. However, tumor-infiltrating T-cell-derived IFN-γ stimulated the expression of Gal-9 on APCs in the HCC microenvironment, suggesting a bidirectional dialogue between DC and T lymphocytes [91]. Moreover, Gal-9 potentiates CD8+ T cell-mediated anti-tumor immunity via TIM-3/Gal-9 interactions between CD8+ T cells and DCs [102]. Further, TIM-3 targeting by specific antibody improves taxane-based chemotherapy in a model of breast cancer through CD8+ T cell-dependent response with increased functional parameters, such as granzyme B expression [95]. In human and mouse tumors, the authors also showed TIM-3 expression predominantly in myeloid cells, suggesting that TIM-3 regulates CD103+ dendritic cell functions [95].

More recently, while DC constitutively expresses Gal-1, Gal-3, Gal-8, and Gal-9, the most abundant lectins are Gal-1 and -3, which are found at the cell surface. In this study, Gal-1 or Gal-3 knockdown in DC enhanced allogeneic T cell responses, mainly mediated by CD4+ T cells, with increased production of IFN-γ and less Il-10, while no effect was observed with Gal-8 knockdown DC [73].

The role of Gal-3 in the control of DC behavior was also demonstrated in PDA patients upon the GM-CSF-secreting allogenic PDA vaccine [78]. Vaccine responder patients developed anti-Gal-3 neutralizing antibodies after immunization. This study also revealed that Gal-3 binds to LAG-3 and promotes suppression of CD8+ T cells through activation of plasmocytoid DC.

Altogether, the arguments above clearly highlight the critical impact of different members of the galectin family in the differentiation and function of dendritic cells.

### 3.3. T Cells

While the role of Gal-1 in the T-cells’ fate is well-documented [103], the impact of Gal-1 on the regulation of lymphocyte gene expression remains poorly evaluated. An interesting study investigated how Gal-1 affects the level of transcription factor expression to understand fundamental parameters involved in the differentiation of CD4+ T cells into T helper cells or regulatory T cells (Treg) [76]. Depending on the Gal-1 concentration used, GATA-3 and FOXP3 mRNA are upregulated, supporting the idea that Gal-1 can promote the differentiation of CD4+ T cells toward Th2 or Treg cells according to the microenvironment conditions. More importantly, human studies confirmed these results. Indeed, Gal-1 promotes T-cell death and mediates T cell-mediated tumor immune escape, inducing a bias in the production of cytokines and a Th2 phenotype in patients with advanced-stage cutaneous T-cell lymphoma [75]. Several years later, another group revealed the negative role of Gal-1 (through its interaction with CD69) on the differentiation and function of Th17 cells, effects that depend on the activation of the Jak3/Stat5 inhibitory pathway [69].

As we described earlier, not only Gal-1, but also Gal-9 could induce T cell death. Linda Baum’s lab showed in 2008 that these two galectins (Gal-1 and -9) require different carbohydrate ligands to kill T cells and also utilize different intracellular death pathways. Gal-9 (but not Gal-1)-mediated T cell death was blocked by intracellular Bcl-2, while Gal-1 (but not Gal-9)-mediated T cell death was prevented by intracellular Gal-3 [74]. Besides, extracellular Gal-3 binding to the surface of T cells alters membrane organization and the formation of an immunological synapse. Its pentavalent capacity allows Gal-3 to interact specifically with different membrane proteins and lipids, influencing endocytosis, trafficking, and T cell receptor signaling (reviewed in [104]). These results revealed complex lattices that are assembled between galectins and their different receptors to promote T cell death or inhibit T cell function. Moreover, Gal-3 was recently demonstrated to sequester IFN-γ in the extracellular tumor matrix, reducing T cell infiltration in melanoma or breast tumors [87].

A recent review describes the role and the mechanisms involved in galectin-mediated development and the progression of different types of leukemia [98], detailing the Gal-9/TIM-3 interaction on the circulating CD8+ T cells to impair immune system function and providing an ideal environment for the proliferation of leukemic cells. In the case of the liver, known to have a physiological tolerogenic environment, it is interesting to note that, in the viral infection process occurring in this organ, immune regulation via TIM-3 signaling does not require its recognition of Gal-9. In this case, TIM-3 expressed at the surface of CD8+ Treg binds to the alarmin, a high-mobility group box 1 (HMGB-1) [105]. By contrast, in hepatocellular carcinoma (HCC), the Gal-9/TIM-3 interaction is necessary to enhance exhausted T cells and to render inefficient anti-cancer immune response [91,106]. These results suggest that the mechanisms controlling immune cell function in cancer may differ from those involved in other pathophysiological situations.

In breast cancer, anti-TIM-3 antibodies activated TIL with an increase of IL-15-induced proliferation and IFN-γ production, but without the need for IL-2. Consequently, IL-15, in combination with TIM-3 blocking antibodies, could potentially act as an IL-2 alternative in tumor CD8 T cell expansion in vitro, an essential event for adoptive T cell therapy [107]. While some degree of activation of infiltrating anti-tumor T cells was observed in mouse and human cancers, PD1+/TIM-3+/CD8+ T lymphocytes display various degrees of functional exhaustion, as shown in patients with regionally metastatic differentiated thyroid cancer [99]. Therefore, this result suggests that the activation of TIL is rather complex.

### 3.4. Natural Killer Cells

A regulatory role of galectins in the interface of innate and adaptative immunity was demonstrated in infections, suggesting that similar functions could be active in tumor immunity [108]. In this respect, TIM-3 is an inducible human NK cell receptor that enhances IFN-γ production in response to Gal-9 [84]. In this study, CD56 and Gal-9 expressions in colon cancer were correlated and associated with poor prognosis, suggesting a role of Gal-9 in tumor immune escape in these tumors [84]. In human gastrointestinal stromal tumors (GISTs), infiltrated NK cells express TIM-3 in 6/8 (75%) of the GIST tissues, and all GIST tissues with TIM-3+ NK cell infiltration, were positive for Gal-9 expression by tumor cells [85], suggesting interactions between NK cells and tumor cells. Baker and coll. also showed that Gal-1-overexpressed-malignant glioma cells control the natural killer (NK) safeguard against early tumor formation by destroying transformed target cells in a process referred to as NK immune surveillance [77] (see also [109,110] for galectins and immunotherapy in glioma).

### 3.5. Other Cells

In the last decade, polymorphonuclear neutrophils have been identified to accomplish essential functions in cancer progression [111]. Studies in infectious diseases revealed galectins as master regulators of neutrophil functions [24]. However, the link between galectins and neutrophils in cancer remains poorly documented and should be investigated in more details. Galectins also regulate angiogenesis and platelet behavior to keep the normal functions of the vascular system [34,112]. A recent study shows that Gal-3 interaction with platelet glycoprotein VI in the colon and breast cancer model promotes metastasis [80]. Altogether, these results strongly suggest the importance of studying the impact of galectins as non-canonical regulators of platelet function in cancer. Interestingly, high levels of Gal-9 were detected in blood γδ T cells from PDA patients when compared with those γδ T cells from healthy individuals [94]. However, the relevance of such a description remains to be evaluated in more depth.

To conclude, it is now clear that galectins as ligands have different receptors on hematopoietic cells, each being able to regulate their function, making the scenario much more complex than initially thought. As an example, Gal-9 also binds to 4-1BB via a site distinct from the binding site of the natural ligand (TIM-3), facilitating 4-1BB aggregation, signaling, and activity in T cells, DC, and NK cells [113]. As a consequence, many mechanisms still need to be understood depending on the organ and their pathophysiological conditions [114].

## 4. Role of Galectins on Cancer Prognosis/Diagnosis

While galectins are active players in tumor immune escape, their expression signature could serve as prognosis/diagnosis biomarkers [55] (Table 2, Table 3, Table 4 and Table 5, and Appendix A). In prostate cancer, our study involving patients (naïve of any treatment) revealed a highly regulated expression of galectins during the disease progression [86]. Three essential galectins, including Gal-1, Gal-3, and Gal-8, were detected on patient samples at all stages of the disease by immunohistochemistry [86]. Gal-1 is a unique member that increases its expression during prostate cancer evolution. Gal-8, formerly called prostate cancer tumor antigen-1, or PCTA-1, is strongly expressed during all stages of the disease; its high and constant expression is indispensable for metastatic properties of prostate tumor cells [43]. Finally, Gal-3 expression progressively decreases until a complete shutdown in advanced prostate cancer patients [86]. This regulated signature could serve as prognostic markers to classify prostate cancer patients [115]. The same approach was used for HNSCC, where the authors analyzed the expression of Gal-1, -3, and -9 and their clinical significance [68]. Galectin expression was scored in tumor cells, infiltrating immune cells, and stromal cells in HNSCC (*n* = 160). While Gal-1 and -9 are identified in tumor cells of 11% of the patients, Gal-3 is expressed in the majority of them (84%). The authors concluded that Gal-1 is a poor predictor of survival and correlates with an invasive outcome, and Gal-9 expression could serve as an indicator of improved survival. Thus, Gal-9 seems to mark a beneficial response, while Gal-1 marks a more aggressive evolution. In the same study, tumor invasion was inversely correlated with Gal-3 expression by tumor cells.

The scenario is more straightforward for some types of cancers than others. For instance, thyroid cancers are Gal-3 positive, while this lectin is absent in normal and benign tissues; consequently, Gal-3 detection could help to improve the diagnosis of thyroid cancer (as reviewed in [39,116]). In PDA, blood Gal-9 levels can serve as a new biomarker because serum concentration of Gal-9 was able to discriminate PDA from benign pancreatic disease and healthy individuals [94]. However, the scenario is more complicated in most of the cancer types as these lectins can also be expressed under physiologic contexts.

Interestingly, antibodies against galectins could arise concomitantly with effective anti-cancer therapy. Indeed, in patients with metastatic melanoma, an anti-CTLA-4 treatment in combination with bevacizumab (an anti-VEGF monoclonal antibody) elicits humoral immunity to Gal-3 and Gal-1; those bi-therapy-treated metastatic patients have improved OS [117]. These results could indicate that the neutralization of these galectins may influence the tumorigenic process. Moreover, circulating Gal-3 may potentially have a prognostic and predictive value for immune checkpoint therapy.

Prostate cancer is one of the most refractory diseases for ICP therapy. However, Sipuleucel-T (DC-based vaccine) is the only immunotherapy authorized by the Food and Drug Administration (FDA) for metastatic and non-symptomatic prostate cancer patients. Remarkably, in patients from IMPACT and ProACT clinical trials, humoral responses (e.g., IgG) against the prostate specific antigen (PSA) and Gal-3 were associated with improved OS [118]. Moreover, we recently demonstrated the essential role of Gal-3 in the establishment of immune tolerance in a mouse prostate cancer model. We showed that the specific targeting of this particular galectin in tumor cells is enough to render the vaccine immunotherapy efficient, with long-term protection against cancer recurrence [119]. These results highlight Gal-3 as an excellent prognosis marker for immunotherapy responders and a potential target when combined with a therapeutic vaccine to benefit prostate and other Gal-3-dependent cancer patients.

As already mentioned, the Gal-9/TIM-3 pathway mediates T-cell senescence, suggesting that this pathway could be a relevant immunotherapeutic target in patients with HBV-associated HCC [91]. The same conclusion applies to gastric cancer [96,120]. In this study, TIM-3, Gal-9, CD3, CD8, and FOXP3 were immunostained in Tissue microarrays (TMA) (*n* = 587); such immunophenotypes were then correlated with clinicopathological and prognosis data. The results demonstrated that TIM-3 was mainly expressed by immune cells, with minimal expression in gastric cancer cells. Gal-9, as TIM-3 ligand, was significantly overexpressed in tumor cells. TIM-3 is thus negatively associated with patients’ OS, while CD8+ T cell density is an excellent prognostic factor for patients with gastric cancer [96]. In colon cancer, the expressions of Gal-9 and CD56 (NK surface marker) were both correlated and represented a poor prognosis factor through its action in the migration of NK cells toward tumors [84]. Thus, galectins could be used as prognostic biomarkers of cancer progression or treatment response.

## 5. Ongoing Clinical Trials Involving Galectins

From 64 clinical trials related to galectins (updated to 1 March 2020; a list that includes their evaluation as new cancer treatments), a vast majority of these studies (48/64) evaluate galectins as indirect biomarkers for response to drug treatments. From these biomarker studies, 2/48 evaluate Gal-1 serum concentration, 7/48 measure Gal-3 as a classical biomarker of cardiac injury to see whether anticancer treatment induces any type of post-therapy heart failure, and 39/48 evaluate TIM-3 expression on T cells from patients after treatments (Appendix A).

Only 14/64 clinical trials test the effect of galectin inhibitors on tumor progression. Among these, three studies evaluate galectin-specific antibodies or carbohydrate compounds (selective or not for only one galectin member). The most used antibodies are TSR-022 (also called HAVCR2) and MBG453, which are two different monoclonal antibodies against TIM-3, as well as RO7121661, a bispecific TIM-3/PD-1 antibody as the most recent strategy. There also exist galectins inhibitors with a carbohydrate nature. For instance, GR-MD-02 and GM-CT-01 bind to the CRD of galectins, or compounds that specifically inhibit one galectin like the modified citrus pectin (MCP) that tightly binds to Gal-3; or OTX008, a selective small-molecule inhibitor of Gal-1 (Appendix A). While several groups nicely demonstrated the role of Gal-1 in controlling angiogenesis and the functions of immune cells in various tumor types, a small number of clinical trials targeting this galectin are still ongoing. We can thus wonder if the targeting of Gal-1 (a pleiotropic and multifunctional protein) generated more adverse side effects than antitumor benefits in the patients. The lack of published information compiling clinical trial results does not help to answer this fundamental question.

As aforementioned, two metastatic melanoma post-analyses of the clinical trial (#NCT00790010) showed elicitation of humoral responses against galectins (e.g., Gal-1 or Gal-3) in patients with successful combinatorial therapeutic strategies [117,121]. Indeed, tumor endothelial activation and immune cell infiltration were associated with favorable clinical outcomes in patients treated by a combination of anti-VEGF (bevacizumab) with anti-CTLA-4 (ipilimumab). This study revealed the essential role of the elicited humoral immune response against Gal-1 in 37.2% of treated patients, those who had improved OS. This clinical result suggests that immunity against Gal-1 may contribute to the success of anti-angiogenic and anti-ICP strategies [121]. In addition, the same combinatorial ICP treatment induced increased titers of antibodies against Gal-3 in one-third of patients, and this humoral response was associated with a favorable clinical outcome [117]. Furthermore, circulating levels of Gal-3 before any therapy could have a prognostic and predictive value for immune checkpoint therapies [117]. 

Several clinical trials involving the analysis of Gal-1, Gal-3, and TIM-3, as the most represented prognostic parameters, demonstrated the relevance of galectins as predictive biomarkers for response to ICP immunotherapy ((Appendix A) and nicely documented in [109,110]). Several antitumor therapies reveal the link between ICP immunotherapies and galectins like Gal-9/TIM-3 for several cancers. In HCC, tumor-associated antigen-specific T cells isolated from human HCC tissues show up-regulation of PD-1, TIM-3, and LAG3, which inhibit the functions of activated TIL. Moreover, antibodies against PD-L1, TIM-3, or LAG3 restore responses of HCC-derived T cells to tumor antigens, and combinations of these antibodies have additive effects, suggesting that blocking those signaling pathways might benefit patients with primary liver cancer [122]. Another group also identified that a poor HCC-specific survival is associated with a low expression of PD-L1 and Gal-9, and a small number of CD8+ TIL [123]. As a result, intra-tumoral expression of these immune inhibiting molecules may be a good predictor of HCC mortality.

The relation between galectins and the immune system in HNSCC was previously reviewed by Kiss lab in 2007 [124]. More recently, ICP immunotherapy failure in this type of cancer was explained by the absence of T cell migration into the tumors, a process that involves the expression of PD-L1 and Gal-9 at the cell surface of the tumor endothelium [51]. In the same way, analysis of peripheral lymphoid and myeloid cells in patients with metastatic non-small cell lung cancer (NSCLC) that are resistant to the anti-PD-1 therapy revealed the accumulation of TIM-3-positive lymphoid cells and Gal-9-expressing monocytic MDSC. These results show that resistance to PD-1 blockade implicates Gal-9/TIM-3 interaction [93].

In multiple myeloma (MM), myeloma cells enhance the function of osteoclasts, which are known to be responsible for bone lesions. Gal-9 expressed by osteoclasts induces the apoptosis of T cells while sparing monocytes and MM cells. Moreover, anti-PD-L1 and anti-IDO therapies overcome the osteoclast induced lysis of cytotoxic T cells and partially explain the benefit of these therapies in MM patients. Anti-Gal-9 therapy in concert with other ICP immunotherapies (PD-L1, IDO) could thus improve anti-MM immunity [92]. These results also strongly support the combinatorial use of ICP immunotherapy with galectin inhibitors.

An in-depth evaluation of current clinical trials demonstrates that only four studies started from 2016 combine ICP therapy (essentially anti-PD-1) with a galectin inhibitor (including anti-Gal-3 or anti-TIM-3) against solid tumors (Table 6), including melanoma, NSCLC, and HNSCC. Surprisingly again, no current combinatory therapy includes inhibitors of Gal-1. Anti-PD-1 immunotherapies in combination with a TIM-3 inhibitor, using TSR-042 (#NCT03708328) or RO7121661, a bispecific antibody against TIM-3 and PD-1 (# NCT04139902), are ongoing for various types of cancers. A former clinical trial in metastatic melanoma used a galectin antagonist (GM-CT-01, essentially a Gal-3 inhibitor) combined with peptide vaccination (#NCT01723813). Moreover, the van der Bruggen’s group showed restoration of the human TIL functions after ex vivo restimulation in the presence of N-acetyllactosamine, a galectin ligand. More interestingly, the same group showed earlier that GCS-100, a polysaccharide specific inhibitor of Gal-3, detached Gal-3 from CD8+ TIL and boosts their IFN-γ secretion and cytotoxic functions and promoted tumor rejection in mice models [125]. Phase 2 of the corresponding clinical trial (#NCT01723813) was terminated in 2015 and revealed higher IFN-γ secretion and cytotoxicity of CD8+ TIL in GCS-100 treated patients. Treating TIL obtained from patients with various cancers only for a few hours resulted in an increased IFN-γ secretion in up to 80% of the samples [126]. Unfortunately, no ongoing study is following these preliminary and encouraging results, highlighting the need to use more specific galectin inhibitors, for instance, GM-RD-02, used in two independent studies (#NCT02117362 and #NCT02575404), instead of using broader spectrum inhibitors like GM-CT-01. Finally, our recent results reveal that Gal-3 is the main inductor of immune tolerance in prostate cancer. Several ongoing or already terminated clinical trials with anti-ICP therapy show prostate cancer as a highly refractory disease for this kind of therapy. Recently, we suggest that the inhibition Gal-3 expressed by the tumor cells promotes strong and long-term immune protective response after tumor resection and vaccination of the animals [119]. Supporting these results, a phase Ib clinical trial (#NCT02117362) combining GR-MD-02 and Ipilimumab (anti-CTLA-4) shows no adverse effects identified owing to GR-MD-02, suggesting that Gal-3 targeting per se does not cause any clinical problem (from Providence Cancer Center and Galectin Therapeutics communications). However, changes in the standard-of-care for melanoma stopped this clinical trial owing to the approbation of Keytruda (Pembrolizumab) instead of Ipilumuman in many patients. For this reason, the following clinical trial (#NCT02575404) using Keytruda plus GR-MD-02 against melanoma and other cancers (oral head and neck, lung) is ongoing (from Providence Cancer Center and Galectin Therapeutics communications). The analysis of a total of 593 patients shows encouraging results. The combinatorial biotherapy showed a 25% rate of complete response for all tested cancers. In melanoma, the response rate is 62.5% compared with the best response of anti-CTLA-4 alone (33%). Interestingly, those patients that respond to combinatorial therapy have reduced MDSC numbers. Unfortunately, this clinical trial revealed significant toxicities and adverse effects in patients treated with Pembrolizumab; GR-MD-02 caused additive side effects (from Providence Cancer Center and Galectin Therapeutics communications). It is interesting to note a high expression of Gal-3 within the tumor microenvironment of aggressive cancers, which correlates with poor survival, for instance, in patients with NSCLC [89]. As we have already mentioned, Gal-3 regulates macrophage differentiation in NSCLC [90], and treatment with GB1107 (a Gal-3 inhibitor) increased tumor M1 macrophage polarization and CD8+ T-cell tumor infiltration. Moreover, GB1107 potentiated the effects of a PD-L1 immune checkpoint inhibitor, inducing higher expression of IFNγ, granzyme B, perforin-1, and Fas ligand molecules. As a consequence, the use of a Gal-3 inhibitor could provide effective and nontoxic therapy in combination with currently used ICP inhibitors or other immunotherapy strategies to boost immune infiltration and responses in NSCLC. In this context, it is interesting to note that not only PD-1 can be targeted in these combinatory strategies because Durvalumab, an anti-PD-L1, in combination with platinum-based chemotherapy, was recently authorized by the FDA as a first-line treatment for patients with extensive-stage small-cell lung cancer. In conclusion, the elucidation of the mechanisms of action of GR-MD-02 or other Gal-3 inhibitors will help to understand the essential role of Gal-3 in the immune tolerance in various types of cancer. It will define the patients who will most benefit from combined therapy.

## 6. Conclusions

Efforts to translate the basic knowledge about galectins to the clinic are currently based on two main goals: first, these proteins are evaluated as biomarkers of prognosis or response to treatments. This type of evaluation predominates in current clinical trials and has proven to be highly relevant. The second goal evaluates Gals as direct targets of therapy; here, the situation is more complex. The initial results seem to indicate that a therapy based only on galectin targeting would not be sufficient. Instead, combinatorial strategies are currently being evaluated, mainly combining the targeting of both galectin and ICP. In this scenario, the design of combinatorial strategies must be guided by mechanistic considerations and tested in preclinical models. The first clinical results seem to indicate that any intervention requires pre-existing antitumor immune responses (or a concomitant spontaneous or induced immunization procedure). Even if an antitumor immune response exists, there is an extreme diversity of immune functions regulated by negative checkpoints with different cellular and kinetics properties (priming/effector function/antibody production/innate system).

As stated in this review, Gals are thus interesting biomolecules to include in combinatory approaches as they are essential regulators at different phases of the immune reactions in the tumor microenvironment. There are various strategies based on Gals-blockade under evaluation using antagonists or inhibitors, antibodies, and other molecular tools that could specifically target each galectin member. Interestingly, the expression of Gals is found at low levels in physiological tissues, except for immune-privileged sites. Thus, it is very likely that their blockade would not induce significant adverse side effects, as demonstrated with Gal-3 inhibitors. Among these clinical trials, it seems that the combination of galectin inhibitors with other immune interventions showed positive results, suggesting that the main function of the Gals is to interfere with the antitumor function of T cells. The details for each protocol (dosage, intervention strategy, and side effects) in clinical trials need to be finely evaluated and probably adjusted according to the obtained results. For this, it seems essential that the results of all clinical trials be published in order to facilitate the reasonable adjustment of anticancer therapy. All the bibliographic arguments exposed in this review support the concept that the transition of cancer immunotherapy from a hopeful vision to standard-of-care treatment for many tumor types requires combinatorial therapies (galectin inhibitors, chemotherapy, and immune checkpoint blockades).

## Figures and Tables

**Figure 1 biomolecules-10-00750-f001:**
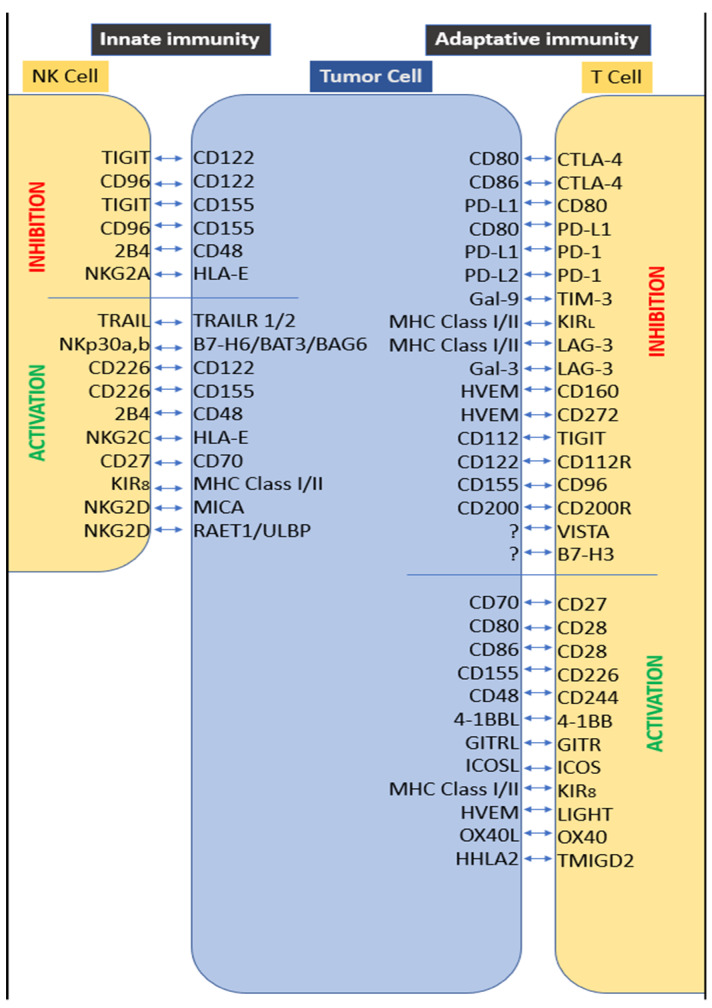
Immune checkpoints in the tumor microenvironment. This figure presents the main immune checkpoints, as membrane-expressed molecules, regulating the immune response in the tumor microenvironment. These immune checkpoints could have activating or inhibiting actions on natural killer cells (NK) or T-cells, influencing the innate or adaptative immunity.

**Table 1 biomolecules-10-00750-t001:** Described functions for the three main galectins (Gals) on immune cells in the tumor microenvironment. NK, natural killer; DC, dendritic cell; MDSC, myeloid-derived suppressor cell.

Cell Types	Myeloid Cells 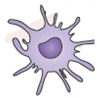	T Lymphocytes 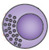	OtherTumor-Associated Cells 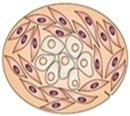
**Galectin-1**	Promotes DC maturation[69,70,71,72,73]Promotes tolerogenic DC [23]Promotes MDSC differentiation[28]	Induces apoptosis[16,74,75]Intrinsic control of T cell function[53,76]Induction of Tregs[37]	Endothelial cells: neoangiogenesis[48,49]NK cells: control of function[77]
**Galectin-3**	Regulates monocyte/macrophage differentiation[75,77]Regulates DC properties[73,78,79]	Induces apoptosis[62]	Platelets: promotes metastasis[80]
**Galectin-9**	Promotes M2 macrophages[81,82,83]Promotes MDSC differentiation[83]Promotes DCmaturation and function[73,79]	Induces apoptosis[74]Regulation of function [73,78,79]	NK cells: control of function[84]

**Table 2 biomolecules-10-00750-t002:** Galectin-1 on immune cells in the tumor microenvironment. OS, overall survival.

Organ/Tumor		Role	Benefit	Prognostic Marker	References
Glioma		Increased in glioma cell, not in host cells	NK immunossurveillance, increase OS		[28,85]
Head and neck squamous cervical cancer	HNSCC	Decreases CD27/CD28	Inversely correlated to OS		[37]
Head and neck squamous cervical cancer	HNSCC	Decreases IFN-γ production	Inversely correlated to OS	Poor prognostic	[51,68]
Hepatocellular carcinoma	HCC	Inhibition of T cell function			[73]
Lymphoma (cutaneous T-cell)		Th2 differentiation			[75]
Pancreatic ductal adenocarcinoma	PDA	Angiogenesis inductor	T-cell infiltration		[50]
Prostate cancer	PCa	Promotes angiogenesis when expressed by tumor cells; enhances immune evasion when expressed by T cells (esp. CD8+ T cells)		Yes	[53,86]

**Table 3 biomolecules-10-00750-t003:** Galectin-3 on immune cells in the tumor microenvironment.

Organ/Tumor		Role	Benefit	Prognostic Marker	References
Breast cancer	BCa	Inhibition of cytotoxic T cell function by IFN-γ sequestration; Gal-3 binds to platelet glycoprotein VI			[80,87]
Colon cancer		Tolerogenic macrophages: M1/M2 differentiation; Gal-3 binds to platelet glycoprotein VI			[80,88]
Head and neck squamous cervical cancer	HNSCC	Increases tumor invasiveness	Decreased OS	Poor prognostic	[68,89]
Hepatocellular carcinoma	HCC	Inhibition of T cell function			[73]
Leukemia (basophilic)		Gal-3 binds to IgE			[55]
Lung carcinoma		Tolerogenic M2 macrophage differentiation and increase of TIL infiltration	Anti-Gal-3 enhances anti-PD-L1 response		[90]
Melanoma		Induces apoptosis of TIL; inhibition of cytotoxic T cell function by IFN-γ sequestration; increase of metastasis	Decreased OS	Yes	[62,63,87]

**Table 4 biomolecules-10-00750-t004:** Galectin-9 on immune cells in the tumor microenvironment.

Ligand	Organ/Tumor		Role	Benefit	Prognostic Marker	References
	Colon cancer		Inhibition of NK infiltration by CD56 on NK interaction with Gal-9		Poor prognostic	[84]
	Gastrointestinal stromal cancer	GIST	Inhibition of NK infiltration		Yes	[85]
	Head and neck squamous cervical cancer	HNSCC	Tolerogenic M2 macrophages differentiation and increase of TIL infiltration; T-cell inhibition		Yes, for treatment response	[68]
	Hepatocellular carcinoma	HCC	Promotes exhausted T cell, and DC differentiation			[91]
	Melanoma		Increases metastasis, and tolerogenic M2 macrophages differentiation	Decreased OS	Yes	[63,82]
	Multiple myeloma	MM	Gal-9 expressed by osteoclasts induces T-cells apoptosis, sparing monocytes, and MM cells.			[92]
	Non-small-cell lung carcinoma (metastatic)	NSCLC	Accumulation of TIM-3 expressed lymphoid cells and Gal-9+ MDSC	Biomarkers of PD-1 blockage resistance	Yes	[93]
**Dectin-1**	Pancreatic ductal adenocarcinoma	PDA	Tolerogenic M2 macrophages differentiation, increase of Gal-9 expression in γδT-cells of PDA patients vs. healthy donors	Biomarkers	Yes	[81,94]
**TIM-3**	Breast cancer	BCa	Expressed by myeloid cells to regulate CD103+ DC functions			[95]
	Gastric cancer				Yes	[96]
	Leukemia (acute myeloid)	AML	Increases mTOR signaling and proangiogenesis		Yes	[97]
	Leukemia (lymphoblastic)		Impaired immune system function			[98]
	Leukemia (stem cell)		Increases MDSCs differentiation	Increased OS		[83]
	Thyroid cancer		Promotes exhausted T cell			[99]

**Table 5 biomolecules-10-00750-t005:** Other galectins on immune cells in the tumor microenvironment.

Galectin	Organ/Tumor		Role	Prognostic Marker	References
**Galectin-7**	Cervical cancer		Increases tumor growth and invasiveness		[67]
**Galectin-8**	Prostate cancer	PCa	Promotes metastasis when expressed by tumor cells	Yes	[43]

**Table 6 biomolecules-10-00750-t006:** Clinical trials involving the combination of immune checkpoint (ICP) and galectin-mediated therapies.

Clinical Trial Numbers	Title	Status	Targets	Tumor Types	Phases	Start Date	End Date
NCT01723813	Peptide Vaccinations Plus GM-CT-01 in Melanoma	Terminated	Galectins + Vaccine	Metastatic Melanoma	Phase 1 Phase 2	April 2012	April 2015
NCT04139902	PD-1 Inhibitor Dostarlimab (TSR-042) vs. Combination of Tim-3 Inhibitor TSR-022 and PD-1 Inhibitor Dostarlimab (TSR-042)	Not yet recruiting	TIM-3 + PD-1	Melanoma	Phase 2	15 March 2020	15 December 2024
NCT03708328	A Dose Escalation and Expansion Study of RO7121661, a PD-1/TIM-3 Bispecific Antibody, in Participants with Advanced and/or Metastatic Solid Tumors	Recruiting	TIM-3 + PD-1	Solid Tumors	Phase 1	15 October 2018	13 September 2022
NCT02575404	GR-MD-02 Plus Pembrolizumab in Melanoma, Non-small Cell Lung Cancer, and Squamous Cell Head and Neck Cancer Patients	Recruiting	Galectins (esp. Gal-3) + PD-1	Various	Phase 1	16 May 2016	October 2021
NCT02117362	Galectin Inhibitor (GR-MD-02) and Ipilimumab in Patients with Metastatic Melanoma	Completed	Galectins (esp. Gal-3) + PD-1	Metastatic Melanoma	Phase 1	8 May 2014	28 November 2018

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
