# Peer review of "Galectins as Checkpoints of the Immune System in Cancers, Their Clinical Relevance, and Implication in Clinical Trials"

_biomolecules, 2020, doi:10.3390/biom10050750_

Round 1

Reviewer 1 Report

Overview

This review article is a very nice overview of the role of galectins, especially cancer associated galectins -1, -3, and -9, with immune responses. Since this article is well-written and very likely to be of wide interest, publication in Biomolecules is appropriate. The focus is on relatively recent publications, which makes this review very useful even in a reasonably crowded, oft reviewed subject area. The reviewers don’t try to be all inclusive. Rather, the article builds effectively to the authors’ main point in the conclusion section, which is that current studies strongly indicate a need for study of combination therapy involving galectin targets. The idea that the main function of galectins may be to interfere with the antitumor function of T cells convincingly arises from the cited studies.

Additional notes

I found Figure 1 to be especially helpful. I

There’s a reference missing on line 140.

Line 181: I believe the verb should be differ rather than differs.

Line 230: Typo. Should say Gal-9/TIM-3

Tables 1 and 2: writing Galectin-1 and galectin-3 vertically and LIGAN/D looks terrible. Omit this column entirely since it is repetitive with the title of the tables.

Tables 3 and 4: reformat so that you don’t have LIGAN/D and sideways information. In table 3, you could eliminate galectin-9 and reformat to shorten one or more of the other columns so that LIGAND is all on one line. In table 4, you could eliminate the BENEFIT column since it is empty and then adjust spacing so that Galectin-7 and Galectin-9 are all on one line.

Line 558: Comforting is not the correct word here. Confirming? Supporting? Corroborating?

Title: Remove “New”. While scientists may only now be realizing that galectins are immune checkpoints that are worth targeting, they are not themselves new checkpoints. A better title would be “Galectins as checkpoints of the immune system in cancers, their clinical relevance, and implication in clinical trials.”

I would have been interested in having some additional information in the galectin-3 section(s) about the focus on CRD (protein/carbohydrate) interactions vs. the N-terminal domain (protein/protein) interactions. How does this change approaches to cancer therapy for galectin-3 relative to galectins -1 and -9?

Author Response

ANSWERS TO THE REVIEWERS’ COMMENTS:

This review article is a very nice overview of the role of galectins, especially cancer associated galectins -1, -3, and -9, with immune responses. Since this article is well-written and very likely to be of wide interest, publication in Biomolecules is appropriate. The focus is on relatively recent publications, which makes this review very useful even in a reasonably crowded, oft reviewed subject area. The reviewers don’t try to be all inclusive. Rather, the article builds effectively to the authors’ main point in the conclusion section, which is that current studies strongly indicate a need for study of combination therapy involving galectin targets. The idea that the main function of galectins may be to interfere with the antitumor function of T cells convincingly arises from the cited studies.

 We thank the reviewer for the thorough review of our manuscript and for bringing to our attention several typographic issues to be corrected in our manuscript.

Additional notes

I found Figure 1 to be especially helpful.

We thank the reviewer for pointing out the necessity of Figure 1.

There’s a reference missing on line 140.

Thanks to reviewer for pointing out this mistake. It was an error in formatting bibliography and no reference was missing. We corrected it in the revised manuscript. LINE 144 (146 in the version with opened change positions)

Line 181: I believe the verb should be differ rather than differs.

We thank the reviewer for pointing out this verbal mistake, which was corrected in the revised manuscript. LINE 184 (186 in the version with opened change positions)

Line 230: Typo. Should say Gal-9/TIM-3.

Done. LINES 252 AND 253 (317 AND 318 in the version with opened change positions)

We agree with the 2 following points and thank the reviewer since it makes the manuscript easier to read. Note that the number of Tables in the revised manuscript changed: Tables 1-5 are now Tables 2-6.

Tables 1 and 2: writing Galectin-1 and galectin-3 vertically and LIGAN/D looks terrible. Omit this column entirely since it is repetitive with the title of the tables.

Tables 3 and 4: reformat so that you don’t have LIGAN/D and sideways information. In table 3, you could eliminate galectin-9 and reformat to shorten one or more of the other columns so that LIGAND is all on one line. In table 4, you could eliminate the BENEFIT column since it is empty and then adjust spacing so that Galectin-7 and Galectin-9 are all on one line.

Line 558: Comforting is not the correct word here. Confirming? Supporting? Corroborating?

DONE. Line 587 (594 in the version with opened change positions)

Title: Remove “New”. While scientists may only now be realizing that galectins are immune checkpoints that are worth targeting, they are not themselves new checkpoints. A better title would be “Galectins as checkpoints of the immune system in cancers, their clinical relevance, and implication in clinical trials.”

We completely agree with the reviewer and change the title as suggested.

I would have been interested in having some additional information in the galectin-3 section(s) about the focus on CRD (protein/carbohydrate) interactions vs. the N-terminal domain (protein/protein) interactions. How does this change approaches to cancer therapy for galectin-3 relative to galectins -1 and -9?

We thank the reviewer for pointing out this particularity of Gal-3 as the unique chimera member of this family. We had omitted this parameter but corrected this omission by arguing this point in the galectin-3 section. LINES 184-203 (187-206 in the version with opened change positions).

Reviewer 2 Report

The manuscript by Compagno and co-workers entitled “Galectins as new Checkpoints….” provides an important view of the clinical trials that consider the biology of Galectins. This is the most important aspect of the paper providing an up-to-date summary of the relevance of Galectin’s for the reader.  Several aspects of the manuscript should be considered though in terms of manuscript structure and content.

1) There are differences in fonts throughout the manuscript, which detracts from readability. 

2) It might be useful for the authors to provide a figure that outlines the important discoveries of Galectin functionality. 

3) The authors should state why they are focusing on the Galectin’s as indicated by the subsections.

4) The subsection headings are off.  After 3.3 T cells comes 2.4 Natural Killer cells.

5) A clear relevance of the cell types discussed by the authors in section 3 should be stated. Maybe a figure emphasizing why these cell types are important as Galectin expressing cells.

6) There are some very long paragraphs and some very short paragraphs – should consider correcting for readability.

7) It would be helpful to expound on the idea of targeting Galectins – either directly (summary of the targeting agents) or as biomarkers.  In the conclusion section the authors seem to imply there are both but hard to decipher in the reading.  A Table or subsection of the therapies that describe mechanism(s) of action that go along with the clinical trials would be most useful and make the manuscript complete.

Author Response

ANSWERS TO THE REVIEWERS’ COMMENTS:

The manuscript by Compagno and co-workers entitled “Galectins as new Checkpoints….” provides an important view of the clinical trials that consider the biology of Galectins. This is the most important aspect of the paper providing an up-to-date summary of the relevance of Galectin’s for the reader.  Several aspects of the manuscript should be considered though in terms of manuscript structure and content.

  We thank the reviewer for the thorough review of our manuscript and for bringing to our attention several typographic issues to be corrected in our manuscript.

1) There are differences in fonts throughout the manuscript, which detracts from readability. 

We have formatted the manuscript to correct such errors.

2) It might be useful for the authors to provide a figure that outlines the important discoveries of Galectin functionality. 

Thanks to the reviewer for the suggestion. Since this review will appear in the special issue “Galectins, their Network and Roles in Infection/immunity/ Tumor Growth Control”, it is very likely that such an illustration about Galectin functions will be part of some other reviews in the same issue. In this scenario and to avoid repetition of information and to keep this review as simple as possible and focused on the clinical relevance of Galectins in cancers, we will ask the editor to confirm whether an illustration of such type has already been proposed by others in this Galectin special issue. In that case, we will only cite that reference as indicated (“additional citation”) in the revised manuscript. However, and in order to include better graphic support that addresses point 5 of this reviewer, we propose a new illustration named Table 1 in the revised manuscript.

Conversely, we believe that the clinical relevance of Galectins in different cancers (the main relevance of this review) is clearly stated in Tables 2, 3, 4 and 5.

3) The authors should state why they are focusing on the Galectin’s as indicated by the subsections.

We thank the reviewer for pointing out the necessity to justify why our manuscript focus on some galectins and not in the others. We added a statement in the revised manuscript. LINES 134-138 (135-139 in the version with opened change positions)

 4) The subsection headings are off.  After 3.3 T cells comes 2.4 Natural Killer cells.

Corrected. LINES 417 AND 430 (421 AND 434 in the version with opened change positions)

5) A clear relevance of the cell types discussed by the authors in section 3 should be stated. Maybe a figure emphasizing why these cell types are important as Galectin expressing cells.

Thanks to the reviewer for this suggestion that improves the comprehension of the text. As previously stated in point 2, a new table (Table 1) is included in the revised manuscript.

6) There are some very long paragraphs and some very short paragraphs – should consider correcting for readability.

Thanks to the reviewer for this note. After revising the entire manuscript, the analysis revealed that the final manuscript is equilibrated in terms of length of each paragraph or respect the relative importance in the bibliographic data of each section. Correcting this format would change the importance of each part by not respecting this bibliographic or existing new data parameter.

7) It would be helpful to expound on the idea of targeting Galectins – either directly (summary of the targeting agents) or as biomarkers.  In the conclusion section the authors seem to imply there are both but hard to decipher in the reading.  A Table or subsection of the therapies that describe mechanism(s) of action that go along with the clinical trials would be most useful and make the manuscript complete.

Thanks to the reviewer for this suggestion that improves the clarity of the message. In the revised manuscript, we have modified the Conclusions section in order to clearly define the two objectives of the clinical uses in which galectins are currently evaluated.

Reviewer 3 Report

The paper is a thorough review of existing knowledge about galectins related to the immune system and cancer. 

It is not qualified as a meta-analysis, but does not claim to be that. 

The reference list is well updated, with the recent as well as the historical references. 

The language must be moderately revised. That might be done in-house. Totally this paper deserves publication as an update of existing knowledge. 

Author Response

ANSWERS TO THE REVIEWERS’ COMMENTS:

The paper is a thorough review of existing knowledge about galectins related to the immune system and cancer. 

It is not qualified as a meta-analysis, but does not claim to be that. 

The reference list is well updated, with the recent as well as the historical references. 

The language must be moderately revised. That might be done in-house. Totally this paper deserves publication as an update of existing knowledge. 

 We thank the reviewer for the thorough review of our manuscript, and two native English speakers had proofread the revised manuscript.